# 'Power plays plus push': experts' insights into the development and implementation of active tuberculosis case-finding policies globally, a qualitative study

Olivia Biermann ,[1] Salla Atkins,[1,2] Knut Lönnroth,[1] Maxine Caws,[3,4] Kerri Viney[1,5]

For numbered affiliations see end of article.

**Correspondence to**
Olivia Biermann;
olivia.biermann@ki.se

## ABSTRACT

**Objective** To explore experts' views on factors influencing national and global active case-finding (ACF) policy development and implementation, and the use of evidence in these processes.

**Design** This is an exploratory study based on semistructured expert interviews. Framework analysis was applied.

**Participants** The study involved a purposive sample of 39 experts from international, non-governmental and non-profit organisations, funders, government institutions, international societies, think tanks, universities and research institutions worldwide.

**Results** This study highlighted the perceived need among experts for different types of evidence for ACF policy development and implementation, and for stakeholder engagement including researchers and policymakers to foster evidence use. Interviewees stressed the influence of government, donor and non-governmental stakeholders in ACF policy development. Such key stakeholders also influence ACF policy implementation, in addition to available systems and processes in a given health system, and implementers' motivation and incentives. According to the interviewees, the World Health Organization (WHO) guidelines for systematic screening face the innate challenge of providing guidance to countries across the broad area of ACF in terms of target groups, settings and screening algorithms. The guidelines could be improved by focusing on what *should* be done rather than what *can* be done in ACF, and by providing *howto* examples. Leadership, integration into health systems and long-term financing are key for ACF to be sustainable.

**Conclusions** We provide new insights into ACF policy processes globally, particularly regarding facilitators for and barriers to ACF policy development, evidence need and use, and donor organisations' influence. According to expert participants, national and global ACF policy development and implementation can be improved by broadening stakeholder engagement. Meanwhile, using diverse evidence to inform ACF policy development and implementation could mitigate the *'power plays plus push'* that might otherwise disrupt and mislead these policy processes.

### Strengths and limitations of this study

► Expert interviews were able to elicit a unique insight into active case-finding (ACF) policy development and implementation.
► Expert interviews filled knowledge gaps regarding factors influencing ACF policy development, donors' influence and evidence use in ACF policy processes.
► The number and diverse range of experts involved increase the study's trustworthiness and confirmability.
► Women and interviewees from low- and middle-income countries were under-represented in the study, potentially limiting the transferability of the results.
► We did not systematically conduct analyses by stakeholder group but described the patterns we observed and highlighted the affiliations of interviewees quoted.

## BACKGROUND

Tuberculosis (TB) is a major global health emergency, especially in low- and middle-income countries. TB is curable and preventable. Still, it remains the leading cause of death from a single infectious agent and one of the top 10 causes of death worldwide.[1] In 2019, the estimated incident TB cases and those notified globally resulted in a difference of 3 million cases, reflecting a combination of under-reporting of detected TB cases and underdiagnosis, specifically in countries with major financial and geographic barriers to accessing care.[1] Many people with TB are diagnosed only after long delays,[2–4] causing increased morbidity, much suffering and economic hardship, and sustaining transmission.[1]

The World Health Organization (WHO) End TB Strategy[5] was endorsed by member states at the World Health Assembly in 2014,

while the United Nations (UN) Sustainable Development Goals[6] were adopted in 2015. Both are aimed at ending the global TB epidemic. Subsequently, there has been increasing international attention on TB. In 2017, the Global Ministerial Conference on Ending TB in the Sustainable Development Era took place in Russia, with the aim of accelerating implementation of the End TB Strategy.[7] In 2018, the UN held the first-ever General Assembly high-level meeting on TB in New York, which endorsed a political declaration to speed up progress towards ending TB. This declaration was adopted by the General Assembly on 10 October 2018.[8] Both the Global Ministerial Conference and the General Assembly re-emphasised the importance of active case-finding (ACF).

Ending TB will require intensified activity to increase TB case detection.[5] One strategy for increased TB case detection is systematic screening, which is defined by the WHO as the 'systematic identification of people with suspected active TB, in a predetermined target group, using tests, examinations or other procedures that can be applied rapidly'.[9] ACF is synonymous with systematic screening for active TB, although it usually implies screening outside of health facilities. ACF is mostly provider initiated. It may target people who do not seek appropriate healthcare because they: do not have or recognise symptoms, do not perceive that they have a health problem requiring medical attention, or face barriers in accessing appropriate care.[9]

ACF has been implemented for decades primarily in high-income countries, starting with mass screening campaigns in the general population in the 1950s and 1960s, then moving towards specific risk populations in recent decades, such as migrants from high-incidence countries and prison populations.[10 11] In low- and middle-income countries, the interest in ACF has increased in recent years, mainly as a response to a sustained case detection gap documented in TB prevalence surveys, annual Global TB Reports produced by WHO[1] and the development of new WHO guidelines on systematic screening.[9]

Questions remain about both *if* ACF in general is worthwhile, as well as *how* to best develop and implement ACF in a given context as a synergistic, rather than parallel structure to the given health system. The evidence base is weak concerning the benefits and cost-effectiveness of ACF on both individual and community levels and how these vary between target risk groups.[12] However, potential benefits of ACF for patients include reduced morbidity, mortality and socioeconomic consequences due to earlier diagnosis, while society can benefit from TB infection prevention, reduced transmission and a reduced burden of TB.[9] There is some evidence that TB screening in high-risk groups can significantly increase TB case notifications.[13–15] Yet, from the health system perspective screening can be costly and lead to diversion of scarce resources. It can also cause harm to patients, for example, by increasing the risk of false-positive diagnoses, creating an additional financial burden associated with attending screening and follow-up, or increased stigma and discrimination, if not properly targeted and implemented.[16]

The potential benefits and challenges of ACF need to be carefully balanced when designing and implementing ACF. Given the relatively weak evidence base for ACF, related policy development and implementation processes rely on stakeholders' tacit knowledge, values and preferences. Yet, little is known about the latter, which potentially impact the development and implementation of national and global ACF policies. The aim of this study was to explore the views of experts on the factors that influence ACF policy development and implementation, and their views of the use of evidence in these processes.

## METHODS

This was an exploratory study based on semistructured expert interviews.[17] The research team used the Consolidated Criteria for Reporting Qualitative Research checklist[18] to report the study (online supplementary file 1).

OB is a doctoral student in public health sciences focusing on ACF. She has experience in qualitative research. The multidisciplinary research team consisting of a medical doctor, an epidemiologist, a microbiologist and a social scientist were involved in this study to ensure different viewpoints were included on ACF policy development and implementation.

### Recruitment and sample selection

The interviewees were purposively sampled to include stakeholders involved in ACF policy development and implementation based at international (n=16), non-governmental (n=2) and non-profit organisations (n=2), funders (n=4), government institutions (n=2), international societies (such as the International Society of Travel Medicine, but in the TB field) (n=2), think tanks (n=1), universities (n=6) and research institutions (n=3), as well as one independent consultant. The research team compiled the initial list of interviewees based on knowledge of networks of experts and on the published scientific literature. The list was discussed with, expanded and verified by two independent experts in the field.

The primary investigator (OB) contacted 50 individuals via email. Of these, two suggested that their colleagues be interviewed instead, eight did not reply and one declined participation due to lack of time and interest. Seven of the 11 people (64%) who declined participation were female. Table 1 provides an overview of the 39 participants who agreed to participate, their sex, professional affiliation and country where they are currently based, classified according to the World Bank.[19] In the Results section, we have used quotes from interviewees across all country income levels to increase the dependability of the results.[20] Moreover, where possible in the results, we have tried to reflect all participants' voices.

### Data collection

OB collected the data between February and May 2018 through semistructured interviews via the phone or in

**Table 1** Participants and their background information (in chronological order)

| ID | Sex | Affiliation | Country classification according to the World Bank[19] |
|---|---|---|---|
| 1 | Male | University | High-income country |
| 2 | Male | International organisation | Low-income country |
| 3 | Male | Government institution | Low-income country |
| 4 | Male | International organisation | Low-income country |
| 5 | Male | Government institution | Low-income country |
| 6 | Male | International organisation | Low-income country |
| 7 | Male | Non-governmental organisation | Low-income country |
| 8 | Male | Non-governmental organisation | Low-income country |
| 9 | Female | Research institution | High-income country |
| 10 | Male | International organisation | High-income country |
| 11 | Male | International organisation | High-income country |
| 12 | Male | Research institution | High-income country |
| 13 | Female | Non-profit organisation | Upper middle-income country |
| 14 | Female | International society | Lower middle-income country |
| 15 | Male | Funder | High-income country |
| 16 | Male | International organisation | High-income country |
| 17 | Female | International organisation | High-income country |
| 18 | Male | Research institution | High-income country |
| 19 | Male | International organisation | High-income country |
| 20 | Male | University | High-income country |
| 21 | Male | University | High-income country |
| 22 | Male | International society | High-income country |
| 23 | Female | Think tank | High-income country |
| 24 | Female | International organisation | High-income country |
| 25 | Male | International organisation | High-income country |
| 26 | Male | International organisation | High-income country |
| 27 | Male | Independent consultant | Lower middle-income country |
| 28 | Male | International organisation | High-income country |
| 29 | Male | International organisation | Lower middle-income country |
| 30 | Male | Funder | High-income country |
| 31 | Male | Funder | Lower middle-income country |
| 32 | Male | University | High-income country |
| 33 | Male | Funder | High-income country |
| 34 | Male | International organisation | Lower middle-income country |
| 35 | Male | International organisation | High-income country |
| 36 | Female | University | Low-income country |
| 37 | Male | University | High-income country |
| 38 | Male | Non-profit organisation | High-income country |
| 39 | Male | International organisation | Upper middle-income country |

person. She developed the interview guides (online supplementary file 2) which MC, KL and KV provided feedback on. The first interview was conducted as a pilot interview after which the guide was revised, making it shorter to focus on the principal topics of interest.

After providing information about the study and obtaining informed written consent, OB asked the interviewees about their experience in developing and/or implementing ACF policies, factors that influenced these policy processes and the use of evidence. The interviews were audio recorded. No repeat interviews were carried

out and no formal field notes were taken. OB conducted interviews aiming to ensure that the sample would hold adequate information power to develop new knowledge.[21] The large number of participants was deemed necessary given the broad aim of the study and that all interviewees had extremely relevant experience related to different aspects of ACF policy development and implementation. This allowed capturing opinions from the diverse range of experts involved in ACF policy development and implementation, but also led to the decision to present parts of the results (on the perceived benefits and risks of ACF) in a separate article to do justice to the breadth and depth of the findings.

Eleven interviews were carried out in person; out of these, eight interviews were conducted during a field visit to Nepal, two during WHO meetings and one during a visit to an international organisation. During the interviews, only OB and the respective interviewee were present. The typical duration of an interview was 30–60 minutes. OB transcribed 10 of the audio-recorded interviews verbatim, while the remaining ones were transcribed by a professional company. The anonymity and confidentiality of the participants were ensured by unique assigned number codes and removing all identifiers except the respondent affiliation in the presentation of the results. OB offered all participants the opportunity to view their transcripts for comments or correction, however, only three participants requested to see the transcripts. No comments or corrections were made by those who chose to view the transcripts.

### Data analysis

OB analysed the qualitative data from the expert interviews with *NVivo V.11* using framework analysis.[17] The data were analysed abductively; defining themes a priori, while allowing for the identification of additional themes based on the data. Using the framework analysis approach as described by Gale *et al*,[22] OB coded all interviews and developed an analytical framework. SA and KV provided comments on the coding, based on which OB revised the codes. The data were then charted into a framework matrix, on which SA and KV provided feedback. OB interpreted the data by writing memos for each study theme, and discussed these with SA, KL and KV. Table 2 provides an example of the coding process.

### Patient and public involvement

The preliminary findings were shared at three different scientific conferences in 2018. The interaction with participants of these events provided unique opportunities for validating the findings. For the presentation of preliminary findings at the World Union Conference on Lung Health, personalised invitations were sent to all 39 interviewees. A few interviewees attended and two provided feedback. As such, the presentation of preliminary findings gave an opportunity for member checking. No direct changes were made based on the validation and member checking, but these processes helped to more critically reflect on the findings. Once published, the results of this study will be reported back to all interviewees. In addition, targeted issue briefs will be developed for researchers and decision-makers in the field. We will also share the results with the public via a video and short messages on social media.

## RESULTS

We generated the following themes from the data: (1) evidence generation and use, (2) factors influencing ACF policy development, (3) factors influencing ACF policy implementation, (4) WHO guidelines on systematic screening, and (5) sustainability of ACF. Table 3 provides an overview of the five main themes and the 16 related codes. The benefits and risks of ACF were additional major themes which will be analysed and discussed in a separate publication. Overall, the interviewees had a wide variety of views on ACF; from ACF being a '*waste basket*' for resources to it being '*common sense*'.

### Theme 1: evidence generation and use

Most interviewees described the evidence on ACF as being relatively limited and emphasised the need to generate different types of evidence to inform ACF policy development and implementation. They stressed the importance of disseminating and exchanging evidence, of the demand for evidence by decision-makers and stakeholder engagement to enable evidence use. Apart from highlighting specific types of evidence, interviewees across the different settings had similar views with regard to this theme.

Interviewees highlighted that a variety of evidence is needed and demanded by decision-makers working on ACF; from effectiveness and health economic evaluations

| Table 2 | Example of the coding process | | | |
|---|---|---|---|---|
| **Interviewee** | **Quote** | **Code** | **Category** | **Theme** |
| I-27, independent consultant in a lower middle-income country | '*So, I think it's the political push that then forces the technocrats to develop policies.*' | Government influencing | Government leadership and commitment | Factors influencing ACF policy development |

ACF, active case-finding.

**Table 3** Summary of major themes and categories related to ACF policy development and implementation

| 1 | Evidence generation and use | 1 | Dissemination and exchange of evidence |
|---|---|---|---|
| | | 2 | Demand for evidence by decision-makers |
| | | 3 | Stakeholder engagement to facilitate evidence use |
| 2 | Factors influencing ACF policy development | 1 | Government leadership and commitment |
| | | 2 | Donor funding |
| | | 3 | Non-governmental organisations' experience |
| 3 | Factors influencing ACF policy implementation | 1 | Human and financial resources |
| | | 2 | Systems, processes and resources to build on |
| | | 3 | Donor funding and related target setting |
| | | 4 | Government power |
| | | 5 | Health workers' motivation and incentives |
| 4 | WHO guidelines on systematic screening | 1 | Positive and negative perceptions |
| | | 2 | Contextualisation of global guidelines locally |
| | | 3 | Suggested improvements |
| 5 | Sustainability of ACF | 1 | Opportunities for sustainability |
| | | 2 | Challenges for sustainability |

ACF, active case-finding.

to implementation and operational research. One interviewee from a university in a high-income country stressed that to demonstrate effectiveness, there is a need '*to do ACF in the context of randomized controlled trials*' (I-32). Another interviewee from a non-governmental organisation (NGO) in a low-income country highlighted the importance of distinguishing clearly where the decisions are being made; be it at the community, district or national level:

I think this is very important, ie what types of evidence you would need to make decisions at various levels (…). What evidence is enough evidence at what level to take the decision. (I-7)

Local evidence was said by many to play a significant role in, for instance, available health and diagnostic facilities, and health workers' capacity and experience in communicating with communities. In particular, evidence from national TB prevalence surveys was described as significant for TB policy development more broadly. Two interviewees from funding organisations in high-income

countries concluded that countries should be encouraged '*to adopt [ACF] policies based on the local evidence and then move forward, rather than waiting for systematic reviews*' (I-30) and '*you should implement enough to figure out what's practicable and what works, and then that should become policy*' (I-15). According to the interviewees, evidence use in ACF policy development and implementation necessitates evidence dissemination and exchange, especially to share unpublished findings. One interviewee from an international organisation highlighted:

Unfortunately, we are [from a low-income country] and we are not very good at publishing. We've got a wealth of experience that is unpublished (…) but it has been presented at several conferences. (I-35)

Depending on the country context, gaps may exist between evidence and policy and/or between policy and practice. As one interviewee from an international organisation in a high-income country pointed out:

Countries are different. As I said, in [that country] (…) from evidence to policy was difficult. But once it [ACF] was inside the policy or even without the policy, they used to easily convert it to practice. But here [in our country] (…) evidence to policy is easier, but policy to practice is more difficult. (I-28)

Interviewees emphasised that researchers should engage with key stakeholders from the beginning of the research process to foster research use in ACF policy development and implementation; stakeholders may include the WHO, the Ministry of Health and the National TB Programme.

'Make sure that you have the right partners from the beginning; partners who are going to take your results and actually do something with them. Because otherwise you are kind of doing it [research on ACF] in a vacuum,' said one interviewee from a university in a high-income country. (I-37)

Moreover, to spark dialogue through stakeholder engagement, an interviewee from an NGO in a low-income country stressed that one must '*create platforms, or you need to use the platforms which are already there*' (I-7). Regular review meetings at subnational and national levels to discuss challenges and successes related to ACF offer one such platform. Overall, evidence use was said to be influenced by *who* is being engaged and by personal contacts which may be '*more important than they should be*', as another interviewee from a university in a low-income country described (I-36).

### Theme 2: factors influencing ACF policy development
According to the interviewees, many different stakeholders influence ACF policy development, specifically governments, donors and NGOs. Interviewees underlined stakeholder involvement as being necessary for policy development and the contextualisation of global policy into local realities. Interviewees did not have any

contradicting views with regard to this theme, but rather highlighted the specific roles of certain stakeholders they thought were most influential in ACF policy development. The leadership, buy-in and commitment of governments and National TB Programmes were described as being vital for ACF policy development and implementation. India was mentioned as a prime example where political push '*forced the technocrats to develop policies and implement them*' (I-27, an independent consultant in a lower middle-income country). Governments make decisions for political reasons or donor incentives, even if these contradict the evidence. One representative from an international organisation in a high-income country highlighted an example of action perceived to be contradicting their view of the evidence:

> Women and children of reproductive age (…) should only be included as part of the passive system not as a priority for ACF ever. But when you talk to NTP [National TB Programme] managers, there is strong political pressure and a perception that donors want them to focus on women and children. (I-24)

Donor organisations such as the Global Fund to Fight AIDS, Tuberculosis and Malaria and the case-finding initiative TB REACH (the latter is coordinated by the Stop TB Partnership and funded largely by Global Affairs Canada) were described as being influential in ACF policy development, for example, TB REACH was said to have '*brought this concept of ACF to the country*' (I-2, interviewee from an international organisation in a low-income country), while the Global Fund '*hold[s] every power to change things and not to change things*' regarding ACF policy development (I-7, representative from an NGO in a low-income country). Likewise, interviewees pointed out that donors' influence was linked to WHO's influence, as donors request countries to adopt WHO guidelines to be eligible for funding:

> 'Why national policymakers are looking mainly at things like WHO documents: because a lot of them get Global Fund money and Global Fund money is often aligned with countries implementing WHO policies,' described an interviewee who is based at a research institution in a high-income country. (I-9)

This observation was shared by another interviewee, from an NGO in a low-income country (I-8). Linking back to the preceding theme on evidence generation and use, TB REACH projects have the potential to generate useful evidence for future policy and practice, as an independent consultant in a lower middle-income country pointed out (I-27). Interviewees said that NGOs are often the 'implementers' of ACF whose years of experience are of great value for ACF policy development and they should therefore be involved in the same, for instance, in policy dialogues with the government and other key stakeholders. One representative of an NGO in a low-income country stated:

> We [NGOs] are the one who really deal with the people (…). We have the evidence. We have the good photographs. We have the data. (…) We are the ones who can influence [ACF policy development]. (I-8)

### Theme 3: factors influencing ACF policy implementation

Interviewees elaborated on available resources, systems and processes within a given health system, donor and government stakeholders, as well as the motivation and incentives for health workers as major factors influencing ACF policy implementation. Interviewees emphasised the role of particular stakeholders, as well as barriers and facilitators they thought were most influential in ACF policy implementation, while no clearly contradictory views on this theme emerged. The implementation and scale-up of ACF policies depend on the availability of financial resources, as many interviewees stressed.

> 'We realized that in a country like [our country], we have great policies. The problem is the implementation. (…) And this is where the support of the development partners, funded through PEPFAR [The President's Emergency Plan For AIDS Relief], have been key to implement these policies, particularly ACF policies,' one interviewee described. (I-39, interviewee from an international organisation in an upper middle-income country)

ACF implementation may '*just stop because [there is] no funding*' (I-29, interviewee from an international organisation in a lower middle-income country). An interviewee from a funding organisation in a high-income country provided a different perspective regarding the funding for ACF by highlighting that '*ACF through government funding can be more difficult than doing it through donor funding*' (I-15). This perspective may inhibit long-term thinking about ACF, as it seems to focus on immediate action to implement rather than sustainability, which is more likely to come with government investment. In addition to limited financial resources, human resource constraints for ACF were highlighted as a major challenge by experts from low-, middle- and high-income countries. These constraints could hinder National TB Programmes in thinking more strategically and ambitiously about how to address TB comprehensively.

The use of existing systems and processes in a given health system was said to be central because '*if you start from scratch, it [ACF] is much more difficult than if there are already things to which you can link*,' as an interviewee from an international organisation in a high-income country pointed out (I-17). Interviewees mentioned that ACF policy implementation can build on experience from existing screening programmes (eg, cervical cancer screening), activities for vulnerable populations (eg, needle exchange programmes), healthcare infrastructure (eg, chest X-ray buses) and already known locations for screening in high-incidence areas and trained human resources (eg, those involved in prevalence surveys). Yet,

pursuing synergies may be challenging due to the fragmentation of activities. In addition, the structure and financing of TB within a health system matters in terms of availability of resources:

'TB has tended to fall into the preventative [arm of the health system] and that has limited the availability for resources,' described an interviewee from a university in a high-income country. (I-32)

Processes including supportive supervision, monitoring and the use of standard operating procedures are critical for ACF policy implementation and are necessary to avoid corruption, interviewees discussed. In one country, the '*whole case-finding system collapsed along with the supervision*' (I-16, interviewee from an international organisation in a high-income country). Moreover, processes that strengthen communication with, engagement of and awareness raising among communities were described as instrumental for ACF policy implementation, for example, to help reduce stigma. One interviewee from a university in a high-income country mentioned how the community '*has started to advocate loudly for ACF services*' (I-20).

Many interviewees underlined that donors influence the implementation of ACF policies in countries with no or insufficient domestic resources. '*The piper will determine what music you play*' (I-35, interviewee from an international organisation in a high-income country), which, again, highlights the power which donor organisations are perceived to have in influencing ACF policy implementation, and the possible resulting lack of a sense of policy ownership in some countries. Donors influence ACF policy implementation by setting targets for their funding recipients and pushing them towards reaching them. One interviewee from an international organisation in a high-income country described:

These targets that countries have set, that donors have set; people are very anxious (…) and that often means the easiest short cut is to do ACF, even if it is a little bit unethical or little bit using low specificity tools, so you have a little bit of over diagnosis. Donors are very comfortable with that. (I-24)

The consequences of implementing ACF under donor pressure are unclear and should be balanced against the unethical nature of inaction on the TB epidemic, but scale-up of inaccurate diagnostic strategies might lead to heightening the potential risks of ACF such as increasing false-positive diagnoses, as the interviewee mentioned.

ACF policy development and implementation depend on '*power plays plus push*', for instance, in a country with no written ACF policy, ACF was still being implemented because the National TB Programme manager was respected and able to push for it (I-29, interviewee from an international organisation in a lower middle-income country). The aforementioned pressure by politicians, donors and WHO may be seen as additional examples of '*power plays plus push*'. Many interviewees highlighted the important role of power dynamics in ACF policy implementation. It seems crucial to be aware of such dynamics, while the use of evidence may help mitigate them. ACF policy implementation is in itself a balancing act, which power imbalances might negatively impact.

The motivation of health workers and volunteers is an important enabler for ACF policy implementation. These 'implementers' can be strongly motivated by their desire to help people, by understanding the benefit of ACF for communities, by receiving feedback on the outcomes of their work (eg, using performance indicators) and/or by feeling ownership of the ACF process, according to the interviewees. Financial and non-financial incentives (eg, salaries, transportation allowances, provision of motorbikes or mobile airtime) have a significant role in motivating health workers and volunteers to implement ACF as an outreach activity, interviewees discussed. Nevertheless, incentives can raise expectations and distort ACF policy implementation in the long term, for example, if government health workers are paid extra as part of an ACF project, they will also expect an extra pay for such activities in the future and for other work; another balancing act. While incentives should be in line with what a country could adopt later, they are often difficult or impossible for governments to sustain, an interviewee said.

### Theme 4: WHO guidelines on systematic screening

This theme focuses on stakeholders' perceptions of the WHO guidelines on systematic screening, the need for their contextualisation and suggestions for improving them. This theme elicited different views among stakeholders, which are described in the following.

The WHO guidelines on systematic screening are perceived positively by many, for instance, as a reference document when planning ACF activities as well as to put ACF on the agenda. Positive perceptions of the guidelines were described by interviewees from different countries, while negative perceptions were only voiced by interviewees in high-income countries. Such negative perceptions included the guidelines being vague, lacking information about the *how-to* of ACF and being unduly negative in terms of mentioning the risk of increasing false-positive diagnoses through ACF. Low-income countries may be more receptive to and reliant on WHO guidelines, while in a middle-income country '*you've got really serious domestic universities providing the formal policy evidence. And the country kind of says "Thanks but no thanks" to outside opinions. They are really driving their own decisions. WHO is really not consulted very much, if at all,*' a representative of a funding organisation in a high-income country described (I-15). An interviewee from an international organisation in a high-income country said:

When you have something that is so broad—and you're talking about ACF which can be so many different things—it's just very hard to have something that works the same way in different countries (…). I

think that's the main shortcoming around the guidance. (I-28)

Interviewees emphasised the necessity of contextualising the WHO guidelines on systematic screening, for example, depending on a country's income level, epidemiology and availability of diagnostic tools. One interviewee from a funding organisation in a high-income country pointed out that '*you just can't be as prescriptive and exact as you are in the more clinical guidelines*' (I-15), which seems like an important observation and reminder about the limitations that ACF policies will always have. According to the interviewees, contextualisation of guidelines can happen in a stepwise approach, for instance, a country pilots the use of a guideline before adopting and adapting it.

Review meetings with WHO and other partners can provide a platform for discussions around guideline adaptation, interviewees said. Yet, countries have faced challenges in contextualisation, for example, WHO recommends using chest X-ray which was too expensive in a country and could thus not be used. In another instance, WHO describes how contacts of an index patient with TB should provide their address, while individuals were hesitant to do so due to the stigma surrounding TB in the country. More support for the contextualisation of guidelines may be needed.

The interviewees suggested that the WHO guidelines for systematic screening[9] must be updated based on new evidence, for instance, evidence from prevalence surveys, gender analyses, studies about specific risk groups (eg, drug users and indigenous populations) and what works and how, with regard to ACF. In this process, WHO should be aware of and avoid conflicts of interest, for example, by ensuring potential conflicts of interests are adequately declared and managed. This comment is in line with what an interviewee previously highlighted about the role of personal contacts to bridge the research–policy gap. These types of biases may undermine the integrity of the process and the resulting quality of guidelines and policies.

Some interviewees lamented that WHO can be paralysed by the need to use the strongest evidence available and suggested that the organisation should consider more programmatic, less scientifically rigorous data. One interviewee from a university in a high-income country described:

> Usually we're relying very heavily on WHO for global policy using the GRADE approach [Grading of Recommendations Assessment, Development and Evaluation was developed for creating summaries of research evidence to help guide health decision-making. It is currently the most widely used tool for evaluating the quality of science, with more than 110 organisations endorsing the method.[23]] with the PICO [P—Patient, Problem or Population; I—Intervention; C—Comparison, Control or Comparator; O—Outcome(s). The PICO process

or framework is a mnemonic used in evidence-based practice to frame and answer a clinical or healthcare-related question. The PICO framework is also used to develop literature search strategies, eg, in systematic reviews.[24]] and all that stuff. I think that's laudable, but sometimes I find that weird, subjected to the tyranny of the great process, and you don't make progress in smaller areas with a paucity of evidence. (I-21)

In addition, interviewees pointed out that WHO recommendations should be based on what *should* be done, not on what *can* be done. For example, countries (not WHO) have to be the ones to decide about their ability to pay for Xpert MTB/RIF as a diagnostic tool. This point of view illustrates a stark contrast to the contextualisation challenges mentioned above, for instance, where the use of X-ray was recommended, but, frustratingly, was unable to be applied in a country as it was not feasible to implement. Moreover, the WHO guidelines could be improved by not only describing the *what*, but the *how* of systematic screening including ACF, many interviewees said. One interviewee from an NGO in a low-income country suggested:

> You can come up with different scenarios: 'If the context is this, then…', 'If the context is that, then…'. (…) Unless guidelines presents [the] 'how' better, (…) it's meaningless. (I-7)

### Theme 5: sustainability of ACF
The sustainability of ACF was a cross-cutting theme in this analysis. Interviewees elaborated on opportunities and challenges related to sustainability. *'TB is not a like smallpox or polio. It's a long-term sustainable (…) matter,'* an independent consultant in a lower middle-income country described. (I-27)

That is, even more perseverance and long-term thinking may be required to end TB. Interviewees expressed similar views and concerns regarding this theme. Interviewees highlighted that the interest in and leadership for ACF through the government and the National TB Programme are important for the sustainability of ACF. Additionally, the sustainability of ACF requires its integration in and funding through the given health system. An interviewee from an international organisation in a low-income country described:

> If this [ACF] were to be sustainable, it should start with the initiation of the NTP [National TB Programme]. (…) It has to be supported, facilitated, monitored. Because it is actually the NTP which later needs to uptake that. (I-2)

Many interviewees highlighted the important role of National TB Programmes. The sustainability of ACF may be restricted in places with frequent government and staff turnover, which makes it difficult to get long-term commitment for ACF from decision-makers, interviewees stressed. Of course, such turnover will affect areas beyond

ACF. Also, ACF cannot be sustainable, if it depends on donor funding. One interviewee from an international organisation in a high-income country summarised the situation as follows:

It [ACF] is difficult to sustain. Most of the activities that have been done for ACF have been project-based. (…) So, the Global Fund comes and says: 'Here is a pot of money for ACF for the next three years.' (…) And then USAID [United States Agency for International Development] comes (…). Or TB REACH (…). And people do it. But that's not a sustainable way of doing this and this should be part and parcel of routine programming. (I-35)

## DISCUSSION

In summary, this study highlighted experts' perceived need for different types of evidence for ACF policy development and implementation, and for stakeholder engagement to foster evidence use. Interviewees stressed the influence of government, donor and NGO stakeholders as influential players in ACF policy development. Such key stakeholders also influence ACF policy implementation, in addition to available systems and processes in a given health system and implementers' motivation and incentives. The WHO guidelines for systematic screening were said to face the innate challenge of covering the broad area of ACF in terms of target groups, settings and screening algorithms. Interviewees suggested that the guidelines could be improved by incorporating new and different types of evidence, by focusing on what *should* be done rather than what *can* be done and by providing examples of the *how* of ACF. Finally, for ACF to be sustainable, interviewees stressed the need for leadership for ACF, its integration into health systems and the transition from donor to government funding.

### Building on a broad evidence base

Interviewees emphasised the need for a variety of evidence, such as impact and economic evaluations, operational and qualitative research. Qualitative evidence has proven essential in developing and implementing health policies including in low- and middle-income countries, for example, to prevent and treat malaria during pregnancy.[25] In the case of ACF, decision-makers may need qualitative evidence on, for instance, factors influencing participation in ACF or the retention of health workers. Likewise, qualitative evidence syntheses have emerged as an important approach to inform national and global health policy development and implementation[26] and could also be useful for improving future ACF policies.

### Making and implementing better ACF policies through stakeholder engagement

Successful ACF policy development and implementation necessitate stakeholder engagement, interviewees highlighted. Stakeholder engagement is an inclusive process essential for achieving legitimate decisions, which are accepted by the population and conducive to effective implementation.[27] Specifically, interviewees stressed the importance of community engagement to enhance the implementation of ACF. Available evidence also shows the importance of community engagement and support for ACF implementation, for example, through collaboration with respected community leaders (ie, chiefs, civic leaders, village elders and counsellors).[28 29] In addition, familiarity with the community[30] and community buy-in[31] as well as community appreciation and respect through the engagement of community health workers was said to be important.[32 33] Stakeholder engagement is also relevant for the development of WHO guidelines at the global level, and their adaptation to the national or subnational levels, where a wide array of stakeholders with diverse sets of values should be involved.[34 35]

### Moving from 'paralyzing' to 'empowering' WHO guidance

The interviewees had many suggestions for improving the WHO guidelines on systematic screening,[9] questioning the appropriateness of only using the GRADE approach in the context of ACF. The WHO guidelines make graded recommendations about screening specific risk groups for TB, including three strong recommendations (screening in household contacts and other close contacts, people living with HIV and current and former workers in workplaces with silica exposure) and four conditional recommendations (screening among prisoners, in people with an untreated fibrotic lesion seen on chest X-ray, in settings where the TB prevalence in the general population is 100/100 000 population or higher, in geographically defined subpopulations with extremely high levels of undetected TB and other subpopulations that have very poor access to healthcare).[9] The conditionality makes decision-making in ACF complex by leaving recommendations open to interpretation. For instance, the conditionality may 'paralyze' decision-makers to move screening outside of health facilities, as ACF in many vulnerable groups is only conditionally recommended. However, despite conditional recommendations and 'low-quality' or 'very low-quality evidence' that all of the WHO's recommendations on systematic screening are based on,[9] decision-makers must still act, either in deciding to implement or taking the decision not to. The Global Fund and TB REACH can provide guidance in interpreting the guidelines. Yet, countries should guarantee that these interpretations and adaptations are based on the local epidemiology, health system capacity, resources, feasibility, effects and economic impact, and so on. This would be paramount in order not to move away from the guidelines' original intention. Ensuring continuous monitoring and evaluation is therefore important.[36] GRADE-Confidence in the Evidence from Reviews of Qualitative Research[37 38] may be a useful resource for future global systematic TB screening guideline development. It has been developed to assess confidence in findings from qualitative evidence syntheses. Additionally,

the GRADE Evidence to Decision Framework for Health System and Public Health Decisions[36] or the WHO-INTEGRATE Evidence to Decision Framework[39] could be valuable to assess evidence for a complex intervention such as ACF.

### Integrating ACF into health systems for sustainability

Interviewees underlined the need to integrate ACF into a given health system for it to be sustainable. Such integration may start with an assessment of the given health system context to understand available structures (eg, infrastructure, budget structure and trained human resources) and processes (eg, supportive supervision and monitoring). Interviewees described these resources as being paramount to link to and build on. The fact that participants highlighted the need for health system integration, which seems to be relevant for any health intervention, may indicate that such integration cannot be taken for granted and/or might not always occur in ACF. It is important to acknowledge that 'integration' may describe a variety of organisational arrangements across different settings.[40] Additionally, in many low-income countries, interventions generally operate through a complex patchwork of arrangements, rather than through totally stand-alone or totally integrated approaches.[41]

To embed ACF into health systems, available systems for outreach and health promotion,[4] laboratory networks[42] and free services[43] have been highlighted. Moreover, given the importance of community health workers for implementing ACF, their integration into the health system has been emphasised.[44 45] Importantly, the collaboration between various actors has been described as key for sustainable ACF implementation. The latter includes collaboration between public health practitioners and clinicians,[46] district TB teams and government health staff,[47] healthcare staff and community health workers.[30 44] Moreover, collaboration between HIV and TB sectors,[48] with laboratory staff[44] and with community organisations[48 49] has been described as important. Government, National TB Programmes, WHO and donors, whose key roles in ACF policy development and implementation have been described by the interviewees, should contribute to the long-term thinking and long-term action related to ACF and towards ending TB. Murphy and Fafard[50] emphasise that only a mix of appropriate evidence, key stakeholders, processes and structures would be a solution for evidence-informed policy development and implementation.

### Future research

Implementation research that sheds light on what works for whom and under which conditions may be particularly helpful to answer some of the *how* questions which our study exposed. Moreover, operational research that uses available local data, for example, on TB notifications, may help inform local decision-making around ACF. Finally, mixed methods studies can help explore the complexity of ACF policy development and implementation in the future, as they have the potential to both increase contextual understanding and reduce biases.

### Strengths and limitations

While the available evidence in this area often focuses on ACF policy implementation,[51] this study fills important knowledge gaps by identifying factors influencing ACF policy development and characterising evidence use in ACF policy development and implementation, from the perspective of experts in the field. Moreover, this study offers an increased understanding of donor organisations' influence on ACF policy processes. The number and diverse range of experts involved in this study, as well as the member checking carried out, increase the study's trustworthiness, including its confirmability and transferability.[20] The transferability of this study's results may be limited given that only a minority of the experts were from low- and middle-income countries (38%; 15 out of 39 experts). Nevertheless, all had working experience from low- and middle-income countries. Seven of the interviews with experts from low- and middle-income countries were conducted with experts from Nepal. Though all of them have different affiliations, their perspectives may be over-represented. The results may furthermore be limited as an even smaller minority were women (18%; 7 out of 39 experts). The gender bias reflects the lack of gender parity in leadership positions in the field of global health.[52] We did not systematically conduct analyses by stakeholder group but described the patterns we observed and highlighted the affiliations of interviewees quoted.

### CONCLUSION

Based on a variety of experts' perspectives, we generated new insights on ACF policy processes, in particular regarding facilitators for and barriers to ACF policy development, evidence need and use, and donor organisations' influence. Still, we know little about *how* to strengthen those facilitators, *how* to overcome those barriers and *how* to strengthen research use. Bringing together these different views creates a more comprehensive picture of ACF policy development and implementation today and indicates ways to strengthen such processes in the future: national and global ACF policy development and implementation can be improved by broadening stakeholder engagement and ownership; from decision-makers at the Ministry of Health to community leaders and members. Meanwhile, using diverse evidence to inform ACF policy development and implementation could mitigate the *'power plays plus push'* that might otherwise disrupt and mislead these policy processes. Our findings complement the existing evidence base and can inform future national and global ACF policy processes.

**Author affiliations**
[1]Global Public Health, Karolinska Institutet, Stockholm, Sweden
[2]New Social Research and Global Health and Development, Faculty of Social Sciences, Tampere University, Tampere, Finland

[3]Department of Clinical Sciences, Liverpool School of Tropical Medicine, Liverpool, UK
[4]Birat Nepal Medical Trust, Kathmandu, Nepal
[5]Research School of Population Health, College of Health and Medicine, Australian National University, Canberra, New South Wales, Australia

**Acknowledgements** The authors thank the interviewees who generously shared their time to participate in the study. The authors also thank Jenny Siméus, writing instructor at Karolinska Institutet University Library, for her valuable feedback in writing this manuscript.

**Contributors** OB, KL, MC and KV conceived the study. OB developed the interview guides, which KL, MC and KV provided feedback on. OB conducted all interviews, coded them and developed an analytical framework. OB revised the coding and the analytical framework based on SA and KV's input. OB charted the data into a framework matrix, which SA and KV provided feedback on. OB interpreted the data writing memos for each study theme, and discussed these with SA, KL and KV. All authors read and approved the final manuscript.

**Funding** This work was supported by the EU-Horizon 2020-funded IMPACT-TB project (grant 733174), from which OB, KL and MC are partly funded. KV is supported by a Sidney Sax Early Career Fellowship from the Australian National Health and Medical Research Council (GNT1121611).

**Competing interests** None declared.

**Patient and public involvement** Patients and/or the public were not involved in the design, or conduct, or reporting, or dissemination plans of this research.

**Patient consent for publication** Not required.

**Ethics approval** This study has been approved by the Swedish Ethical Review Authority (Regionala Etikprövningsnämnden) in Stockholm (reference: 2017/2281-31/2). Participants received information about the study and provided written informed consent.

**Provenance and peer review** Not commissioned; externally peer reviewed.

**Data availability statement** All data relevant to the study are included in the article or uploaded as supplementary information. The data generated and/or analysed in the study are not publicly available due to participant anonymity.

**ORCID iD**
Olivia Biermann http://orcid.org/0000-0002-5978-0211

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
