## [Reviewer comments · BMJ Open]

ARTICLE DETAILS

TITLE (PROVISIONAL)	“Power plays plus push”: Experts’ insights into the development and implementation of active tuberculosis case-finding policies globally, a qualitative study
AUTHORS	Biermann, Olivia; Atkins, Salla; Lönnroth, Knut; Caws, Maxine; Viney, Kerri

VERSION 1 – REVIEW

REVIEWER	Aphra Garner Purkis Kings College London, United Kingdom
REVIEW RETURNED	29-Dec-2019

GENERAL COMMENTS	Overall a really interesting piece and very rich data which supports the results and conclusions drawn from them. Got slightly confused about the use of the acronym NTP which could maybe be used in its longer forms at times as doesn't always seem to work. Line 44- instead of 'the study accentuated' the study highlighted or brought to light Line 56- generated new knowledge seemed like a strange way of saying this generated from the data from the interviewees experiences? Line 84 -instead could change to In 2019, the estimated incident tb cases and those notified globally resulted in a difference of three million cases reflecting..... Line 91- both are aimed at Line 122- when it says economically productive members of society maybe in brackets this could include what these main demographics include (25-34....) Line 147-maybe just me but not sure what an international network is, could be explained further ? Line 164- not sure conversational form is necessary, and also maybe be consistent with line 176 and say either in person or face to face Line 181- unique assigned number codes? Line 225 and 226- words in " " should be in italics if from the interviews Line 239- instead of underlined could be outlined/highlighted Line 248- could change to Local evidence was said by many to play a significant role in for instance available health and diagnostics facilities..... Line 363- should be (e.g. chest X-ray buses) instead of busses Line 397 and 398 is this a direct quotation or a author worded version of what was said ?
--

REVIEWER	Carrie Tudor
-----------------	--------------

	International Council of Nurses, Switzerland
REVIEW RETURNED	19-Jan-2020

GENERAL COMMENTS	Thank you for the opportunity to review this manuscript. Please see comments in attached file and below. Response to Authors Thank you for the opportunity to review this interesting manuscript. Below please find my comments for your consideration.  1. Abstract: The text in the objective, design and participants sections seems awkward and wonder if you may want to consider flushing the text out a bit so that it is a bit more narrative. May also want to add a background section before the objective. I find your objective statement regarding the “perspectives of experts” interesting as experts – especially from universities and from high-income countries often have no involvement or very little involvement in the design and implementation of guidelines and policy at a country level. 2. Page 5 -6. It is interesting that you do not discuss the important role that ACF and screening play in airborne/TB infection prevention and control – especially in high-burden settings. 3. Top of page 7, line 152. You may want to consider adding the following text “The primary investigator (OB...” so that it is clear to the reader who “OB” is and what their role was. 4. In reviewing the COREQ checklist attached – you mention that most information is on page 6 of the manuscript but it seems that there is quite a bit of the information found on other pages. You may consider reviewing and revising as necessary. 5. Page 9, line 173. It seems that you conducted interviews past saturation based on your sentence “OB conducted interviews beyond reaching information power.” Was there a reason for this? Why did you decide to do this? Was there additional information you gained by doing this? You may want to add a sentence or comment as to why you did this and what you gained by doing so. 6. Page 9, line 179. Who transcribed the interviews? Was the transcription done by OB who also conducted the interviews or another person? Please add to text. 7. Page 10, line 183. How many is a “handful”? Please add the number of participants who asked to see their transcripts. 8. Page 10, lines 187 – 190. It seems that this text is required for the COREQ checklist. You may want to consider adding this text into other paragraphs where it can blend better. I think I know what you are trying to say with “...helped elicit the diverse perspectives on ACF policy development and implementation” – but it seems that the point of the research and interviews with participants is what elicited the diverse perspectives on policy and development. May want to consider revising this sentence to something like – “the multidisciplinary research team helped in the design of the study to ensure different viewpoints were included..... I am guessing that is more of what you meant.
--

	9. Page 10, line 193 & 194. Data are plural – “data were” 10. Page 11, lines 206 – 210. Not sure this is relevant. If required, consider simplifying and simply state that the data were shared with participants at an international conference.... A few lines down in the same paragraph – you do not list the number of participants who attended for member checking. You may want to add this. 11. Page 11, line 223 – please spell out numbers < 10 throughout 12. You may want to consider deleting the sentence on page 11, line 223 – 224 regarding the benefits and risks of ACF... - not relevant to this manuscript. 13. Results and quotes – some of the quotes are quite long so you may want to consider indenting them so they stand out a bit more for the reader. Additionally, some of the quotes are a bit awkward in that names of places (countries) or other identifying information is omitted for confidentiality and that makes the quotes a bit awkward. It may actually take away from the intended meaning and impact of the quote. Are there different quotes to use that make the same point with less context removed? 14. Page 22, line 465. What do you mean by this? What are the perceived conflicts of interest? 15. Page 25 “Strengths and limitations” – this section should be moved to page 29 16. Page 25, lines 550 – 551. How do you think the small number of female respondents influenced the findings? How many of those who did not participate in the study women? 17. Pages 25 & 26, lines 552-557. May consider deleting from page 25, end of line 25 “Due to the...” through end of line 557 on page 26. This is not relevant to this paper and can be deleted. 18. Page 27, line 597 is “not” missing at the end of this line between “likely” and “from”? 19. Page 29, line 635. May want to consider changing the section heading to “Conclusion”
--	---

REVIEWER	David Scales Weill Cornell Medical College, United States
REVIEW RETURNED	19-Feb-2020

GENERAL COMMENTS	This is quite a well done study of attitudes and perceptions of key stakeholders (policymakers, government, NGO, etc) perceptions of active case finding policy development and implementation, particularly focusing on tacit knowledge, values and preferences for which there is little literature. The study is well carried out with a significant number of interviews among diverse stakeholders, making thematic saturation likely. Methodologically, it is theoretically grounded with appropriate attention to reflexivity and other characteristics that increase robustness of qualitative research and COREQ items, and the results are interesting and useful for the
--

	field, with implications both for policymaking and future research. The coded themes are presented with key illustrative quotes giving appropriate context to the results and conclusions. There are some weaknesses to the paper, most prominently the potential bias in how stakeholder informants were chosen - via list generation of a network of experts that was then expanded and verified by two independent field experts. The reader knows little about these experts on who so much rests in expanding the list of potential informants, so it would be helpful to know some characteristics on these experts, especially given the acknowledged limitation that the informant interviews lack representation from women and people from LMI countries. This weakness is particularly concerning because at least half (8/15) of the low income country representation was likely generated through one field visit to Nepal, perhaps over-representing one perspective. Reasonable attempts to expand the informant base were made for an exploratory study and the weaknesses are acknowledged and may represent a dearth of diversity in global health leadership positions. Therefore this weakness should not prevent the publication of the paper. Secondly, the paper should describe the informed consent procedure(s), or if none was required.
--	---

VERSION 1 – AUTHOR RESPONSE

#	Reviewers' comments	Authors' answers
	Reviewer 1: Aphra Garner-Purkis	
4	Got slightly confused about the use of the acronym NTP which could maybe used in its longer forms at times as doesn't always seem to work.	We have removed the acronym all together in order to increase the clarity of the text.
5	Line 44- instead of 'the study accentuated' the study highlighted or brought to light	We have revised the text accordingly (line 43).
6	Line 56- generated new knowledge seemed like a strange way of saying this generated from the data from the interviewees experiences?	We have revised the text in line with that (lines 57 and 726): " We generate new knowledge provide new insights into... "
7	Line 84 -instead could change to In 2019, the estimated incident tb cases and those notified globally resulted in a difference of three million cases reflecting.....	We have revised the text, as suggested (lines 92-94).
8	Line 91- both are aimed at	We have revised the text accordingly (line 101).
9	Line 122- when it says economically productive members of society maybe in brackets this could include what these main demographics include (25-34....)	We have deleted this subclause as the information it contained was not essential: "...society can benefit from TB infection prevention, reduced transmission and a reduced burden of TB, which often affects the most economically productive members of a society [9]." (line 132).
10	Line 147-maybe just me but not sure what an international network is, could be explained further ?	We now refer to the "international networks" as " international societies ", hoping that this might more clearly indicate the nature of these two types of organizations (lines 38, 162, 501 and Table 1). Moreover, when first mentioned, we have also added " (such as the International Society of Travel Medicine, but in the TB field) " for clarity (lines 162-

#	Reviewers' comments	Authors' answers
		163).
11	Line 164- not sure conversational form is necessary, and also maybe be consistent with line 176 and say either in person or face to face	We have deleted the addition that the interviews were done "in conversational form" (line 181). Moreover, we revised the text to be consistent by replacing "face-to-face" with "in person" (line 201).
12	Line 181- unique assigned number codes?	We have revised the text, as suggested (line 207).
13	Line 225 and 226- words in " " should be in italics if from the interviews	We have now made sure these words are in italics, as they indeed derive from the interviews (lines 253-254): "Overall, the interviewees had a wide variety of views on ACF; from ACF being a "waste basket" for resources to it being "common sense" ."
14	Line 239- instead of underlined could be outlined/highlighted	We have revised the word accordingly, using "highlighted" instead of "underlined" (line 267).
15	Line 248- could change to Local evidence was said by many to play a significant role in for instance available health and diagnostics facilities.....	We have revised the text, as suggested (line 277).
16	Line 363- should be (e.g. chest X-ray buses) instead of busses	We have corrected the word, as proposed (line 404).
17	Line 397 and 398 is this a direct quotation or a author worded version of what was said ?	We would like to confirm that this is a direct quote from I-29. Reference to I-29 is made in the end of the sentence (lines 443-444).
	Reviewer 2: Carrie Tudor	
18	Abstract: The text in the objective, design and participants sections seems awkward and wonder if you may want to consider flushing the text out a bit so that it is a bit more narrative. May also want to add a background section before the objective.	We have revised the abstract accordingly (lines 28-40). We have also double-checked BMJ Open's guidelines which recommend against including a background section, which is why we have not included one.
19	I find your objective statement regarding the "perspectives of experts" interesting as experts – especially from universities and from high-income countries often have no involvement or very little involvement in the design and implementation of guidelines and policy at a country level.	We would like to confirm that we only interviewed experts who have indeed been actively involved in ACF policy development and implementation processes; at national and/or global level, most of them both. The inclusion of both national and global levels is highlighted in lines 29-30: "To explore experts' views on factors influencing national and global active case-finding (ACF) policy..."
20	Page 5 -6. It is interesting that you do not discuss the important role that ACF and screening play in airborne/TB infection prevention and control – especially in high-burden settings.	Previously, we have only mentioned the potential benefit of "reduced transmission and a reduced burden of TB" (line 132), however, we have now explicitly added "TB infection prevention" among other potential benefits in line 132.
21	Top of page 7, line 152. You may want to consider adding the following text "The primary investigator (OB..." so that it is clear to the reader who "OB" is and what their role was.	We have added this to the text (line 168).
22	In reviewing the COREQ checklist attached – you mention that most information is on page 6 of the manuscript but it seems that there is quite a bit of the information found on other pages. You may consider reviewing and revising as necessary.	We have reviewed and revised the COREQ checklist accordingly (see separate file).
23	Page 9, line 173. It seems that you conducted interviews past saturation	We have now clarified our reason for doing so in the text (lines 191-199): "In order to capture opinions from

#	Reviewers' comments	Authors' answers
	based on your sentence "OB conducted interviews beyond reaching information power." Was there a reason for this? Why did you decide to do this? Was there additional information you gained by doing this? You may want to add a sentence or comment as to why you did this and what you gained by doing so.	the diverse range of experts involved in ACF policy development and implementation, OB conducted interviews beyond reaching information power aiming to ensure that the sample would hold adequate information power to develop new knowledge [21]. The large number of participants was deemed necessary given the broad aim of the study and that all interviewees had extremely relevant experience related to different aspects of ACF policy development and implementation. This allowed capturing opinions from the diverse range of experts involved in ACF policy development and implementation, but also led to the decision to present parts of the results (on the perceived benefits and risks of ACF) in a separate article in order to do justice to the breadth and depth of the findings.
24	Page 9, line 179. Who transcribed the interviews? Was the transcription done by OB who also conducted the interviews or another person? Please add to text.	We have now clarified this in the text (lines 204-206): "OB transcribed 10 of the audio-recorded interviews were transcribed verbatim, while the remaining ones were transcribed by a professional company."
25	Page 10, line 183. How many is a "handful"? Please add the number of participants who asked to see their transcripts.	We have now clarified this in the text (line 210): "a handful three of the participants..."
26	Page 10, lines 187 – 190. It seems that this text is required for the COREQ checklist. You may want to consider adding this text into other paragraphs where it can blend better. I think I know what you are trying to say with "...helped elicit the diverse perspectives on ACF policy development and implementation" – but it seems that the point of the research and interviews with participants is what elicited the diverse perspectives on policy and development. May want to consider revising this sentence to something like – "the multidisciplinary research team helped in the design of the study to ensure different viewpoints were included..... I am guessing that is more of what you meant.	We have moved this text to the beginning of the methods section; mainly to introduce the principal investigator, OB, who we refer to in the subsequent paragraphs. We have also revised the text based on your suggestions, i.e. that the multidisciplinary team was "involved in this study to ensure different viewpoints were included..." (lines 154-157).
27	Page 10, line 193 & 194. Data are plural – "data were"	We have corrected the text accordingly (line 220).
28	Page 11, lines 206 – 210. Not sure this is relevant. If required, consider simplifying and simply state that the data were shared with participants at an international conference.... A few lines down in the same paragraph – you do not list the number of participants who attended for member checking. You may want to add this.	We consider this relevant to show that we have discussed and reflected on our preliminary findings at different occasions. As recommended, we have simplified the text and clarified the number of participants who attended for member-checking below (lines 232-238): "The preliminary findings were shared at three different scientific conferences in 2018, such as the Global Symposium on Health Systems Research, the First Annual Conference on Implementation Science and Scale-up and the World Union Conference on Lung Health. The interaction with participants of these events provided unique opportunities for validating the findings. For the

#	Reviewers' comments	Authors' answers
		presentation of preliminary findings at the World Union Conference on Lung Health , personalized invitations were sent to all 39 interviewees. A few interviewees attended and two provided feedback."
29	Page 11, line 223 – please spell out numbers < 10 throughout	We have revised the number accordingly (line 251), and also double-checked the entire text on this issue.
30	You may want to consider deleting the sentence on page 11, line 223 – 224 regarding the benefits and risks of ACF... - not relevant to this manuscript.	To be in line with item 31 of the COREQ checklist ("Clarity of major themes"), we left this sentence ("The benefits and risks of ACF were additional major themes which will be analyzed and discussed in a separate publication." [lines 251-254]), so the reader would be able to understand how we handled the analysis of this major theme.
31	Results and quotes – some of the quotes are quite long so you may want to consider indenting them so they stand out a bit more for the reader. Additionally, some of the quotes are a bit awkward in that names of places (countries) or other identifying information is omitted for confidentiality and that makes the quotes a bit awkward. It may actually take away from the intended meaning and impact of the quote. Are there different quotes to use that make the same point with less context removed?	We have indented selected quotes, as recommended. In addition, we agree that it is important to include identifying information along the quotes. We have therefore double-checked the quotes but could not find any that did not identify the interviewee's affiliation and country classification; it is either mentioned in brackets just after each quote or in the text. However, we have corrected this misleading sentence (lines 720-721): "We have highlighted the affiliations of interviewees quoted when relevant ."
32	Page 22, line 465. What do you mean by this? What are the perceived conflicts of interest?	We have now clarified this in the text (lines 511-512). "WHO should be aware of and avoid conflicts of interest, e.g. by ensuring potential conflicts of interests are adequately declared and managed."
33	Page 25 "Strengths and limitations" – this section should be moved to page 29	We have moved this section, as suggested (lines 704-721).
34	Page 25, lines 550 – 551. How do you think the small number of female respondents influenced the findings? How many of those who did not participate in the study women?	We have added to methods section (lines 170-171): " Seven of the 11 people (64%) who declined participation were female. " In terms of how the small number of female respondents influenced the findings, we can only speculate that women may have raised different topics, such as gender issues in ACF policy development and implementation and other matters more likely faced by women, based on their own experience. This may be an interesting topic for future studies to address.
35	Pages 25 & 26, lines 552-557. May consider deleting from page 25, end of line 25 "Due to the..." through end of line 557 on page 26. This is not relevant to this paper and can be deleted.	We have deleted the text, as suggested (lines 721-726). We also deleted lines 558-561 for the same reason: " We have also conducted a cross-sectional survey with NTP managers from the 30 high-burden TB countries which will shed light on their views on ACF policy development and implementation, including the sustainability of ACF (manuscript in preparation). "
36	Page 27, line 597 is "not" missing at the end of this line between "likely" and "from"?	We chose to delete this sentence, as the type of information we refer to in the preceding phrase indeed derives from different types of studies, making it less useful to specify just a few (lines 652-653): " This information is most likely from qualitative studies, monitoring and evaluation research and quasi-

#	Reviewers' comments	Authors' answers
		experimental studies.
37	Page 29, line 635. May want to consider changing the section heading to "Conclusion"	We renamed this section "Future research" (line 692) and moved parts of the text under Conclusion below (lines 729-733): "Still, we know little about how to strengthen those facilitators, how to overcome those barriers and how to strengthen research use."
	Reviewer 3: David Scales	
38	The reader knows little about these experts on who so much rests in expanding the list of potential informants, so it would be helpful to know some characteristics on these experts, especially given the acknowledged limitation that the informant interviews lack representation from women and people from LMI countries. This weakness is particularly concerning because at least half (8/15) of the low income country representation was likely generated through one field visit to Nepal, perhaps over-representing one perspective. Reasonable attempts to expand the informant base were made for an exploratory study and the weaknesses are acknowledged and may represent a dearth of diversity in global health leadership positions. Therefore this weakness should not prevent the publication of the paper.	We have previously discussed this issue with all authors and have re-considered it. Based on these discussions, we decided not to include any further characteristics of the interviewees in order not to compromise their anonymity. It is correct that 7/15 interviewees from LMICs were from Nepal. We have now added a comment on this to the limitations section to be fully transparent (lines 715-717): "Seven of the interviews with experts from low- and middle-income countries were conducted with experts from Nepal. Though all of them have different affiliations, their perspectives may be overrepresented."
39	the paper should describe the informed consent procedure(s), or if none was required.	Participants did provide written informed consent. We have now added a sentence to the methods section (line 186): "After providing information about the study and obtaining informed written consent, OB asked the interviewees about their experience..." The consent procedure is also mentioned under the "Ethics approval and consent to participate" and "Consent for publication" sections (lines 909-915).

VERSION 2 – REVIEW

REVIEWER	Aphra Garner-Purkis Kings College London
REVIEW RETURNED	09-Apr-2020
GENERAL COMMENTS	Really interesting paper to read again and really interesting topic. Just a few grammatical things that I have highlighted. Abstract Participants heading-not sure why this heading is participants and then includes about framework analysis as a method? Body of paper

	line 56 a/the knowledge? Line 173 revised, making it shorter (take out by) Line 188 is it an international organisation meeting ? Line 191 full stop after company Line 321 needed [a] before low income country Line 335 available resources..... and resources (repetition not sure if the resources are referring to two different things) Line 339 don't need while before no clearly
--	--

REVIEWER	Carrie Tudor International Council of Nurses, South Africa
REVIEW RETURNED	31-Mar-2020

GENERAL COMMENTS	Thank you for addressing all of the comments. I have no further comments.
---

REVIEWER	David Scales Weill Cornell Medical College
REVIEW RETURNED	20-Mar-2020

GENERAL COMMENTS	My comments from the previous review have been addressed. I recommend acceptance and publication of the manuscript.
---